# Melatonin-Induced Transcriptome Variation of Sweet Potato Under Heat Stress

**DOI:** 10.3390/plants14030430

**Published:** 2025-02-01

**Authors:** Mengzhao Wang, Yang Zhou, Bei Liang, Sunjeet Kumar, Wenjie Zhao, Tianjia Liu, Yongping Li, Guopeng Zhu

**Affiliations:** 1School of Breeding and Multiplication (Sanya Institute of Breeding and Multiplication), Hainan University, Sanya 572025, China; mzwang@hainanu.edu.cn (M.W.); 184224@hainanu.edu.cn (S.K.);; 2Key Laboratory for Quality Regulation of Tropical Horticultural Crops of Hainan Province, School of Tropical Agriculture and Forestry, Hainan University, Haikou 570228, China

**Keywords:** sweet potato, heat stress, melatonin, transcriptome, oxidative stress

## Abstract

Melatonin (MT) has been widely recognized for its ability to mitigate the effects of abiotic stress and regulate plant development. In this study, we investigated the role of exogenous MT in enhancing heat tolerance in sweet potato, with a particular focus on its capacity to alleviate heat stress-induced damage. MT treatment significantly reduced oxidative stress, as evidenced by decreased levels of hydrogen peroxide, superoxide ions, and malondialdehyde (MDA), all of which were elevated under heat stress. To uncover the underlying mechanisms, RNA sequencing was performed on three experimental groups: control (CK), heat stress alone (HS), and MT pre-treatment followed by heat stress (MH). A total of 3491, 3280, and 1171 differentially expressed genes (DEGs) were identified in the CK vs. HS, CK vs. MH, and HS vs. MH comparisons, respectively. MT treatment notably modulated the expression of genes involved in redox regulation and nicotinate and nicotinamide metabolism. Moreover, MT enhanced the expression of genes associated with key signaling pathways, including mitogen-activated protein kinases (MPK3) and plant hormone signal transduction components, such as ethylene response factor (ERF). These findings offer novel insights into the mechanisms by which exogenous MT enhances heat tolerance in sweet potato, highlighting its role in regulating antioxidant systems, metabolic pathways, and hormone signaling. This study presents valuable strategies for improving crop resilience to heat stress.

## 1. Introduction

Heat stress, caused by elevated temperatures, poses a significant threat to agriculture worldwide [1]. Transitory or constantly high temperatures can lead to an array of structural, physiological, and biochemical alterations in plants that disrupt growth and development, often resulting in substantial yield losses. This stress affects plants at various developmental stages, from seed germination to maturity, with each stage exhibiting unique sensitivity to heat [1,2]. For instance, extreme temperatures during germination can delay or even halt the process entirely, while heat stress in later stages can impair key processes such as photosynthesis, respiration, and water balance, as well as compromise membrane integrity and hormone regulation [3]. These stress-induced changes trigger complex protective responses in plants, including the upregulation of heat shock proteins, synthesis of other stress-related proteins, and production of reactive oxygen species (ROS) [2]. While ROS act as signaling molecules at low concentrations, triggering protective responses, excessive ROS accumulation can cause oxidative damage to cellular components. Effective detoxification of ROS is therefore critical for stress tolerance. Developing thermotolerant crop varieties through genetic and biotechnological approaches is critical to mitigating these effects. However, achieving this goal requires an in-depth understanding of plant physiological and molecular responses to heat, the mechanisms that underpin thermotolerance, and potential strategies for enhancing crop resilience to elevated temperatures.

Melatonin has gained attention as a multifunctional molecule involved in plant stress tolerance [4,5]. Melatonin, a well-known regulator of circadian rhythms in animals, also plays critical roles in plants under abiotic stress conditions, such as drought, salinity, and temperature extremes [6]. One of its primary functions is the detoxification of ROS, which are generated excessively under heat stress. Through its antioxidant capacity, melatonin helps maintain cellular homeostasis by scavenging ROS, protecting cellular structures, and modulating stress-responsive genes [7]. Melatonin’s involvement in plant stress tolerance primarily involves ROS scavenging. It has been shown to regulate the expression of heat shock proteins and activate antioxidant enzymes, which collectively contribute to maintaining cellular integrity, including stabilizing cell membranes under heat stress conditions [6]. These functions enable plants to cope with environmental stress by maintaining cellular function and enhancing stress recovery. Transcriptomic studies have revealed that melatonin treatment leads to significant changes in gene expression patterns, particularly genes related to heat stress responses, suggesting that melatonin has a regulatory role in the heat stress response at the molecular level [8,9].

However, despite these insights, there remains a gap in understanding how melatonin affects transcriptome-level changes specifically in sweet potato seedlings under heat stress. Most studies have focused on model species or other economically significant crops, leaving the response in sweet potato largely unexplored. In previous studies, we examined the effect of the antioxidant defense system, osmotic homeostasis, and stomatal traits in response to exogenous melatonin on sweet potato at the physiological level [10]. Our findings revealed that heat stress significantly reduced both shoot and root fresh weight, leading to oxidative stress, and caused a decrease in chlorophyll concentration, photosystem II efficiency (Fv/Fm), and gaseous exchange. However, pre-treatment with melatonin improved heat stress tolerance in sweet potato by increasing antioxidant enzyme activities and a rise in non-enzymatic antioxidants and osmo-protectants. In this study, we aimed to (1) How does exogenous melatonin modulate the transcriptomic responses of sweet potato under heat stress? (2) What are the key metabolic pathways and regulatory networks influenced by melatonin to enhance heat tolerance? Based on these questions, we hypothesize that (1) exogenous melatonin enhances heat tolerance in sweet potato by modulating stress-responsive genes involved in antioxidant defense and hormonal signaling pathways and (2) melatonin treatment regulates specific metabolic pathways, such as nicotinate and nicotinamide metabolism, which contribute to improved stress resilience.

## 2. Results

### 2.1. Identification of Differential Expression Genes (DEGs)

To uncover the potential regulatory mechanisms underlying melatonin-mediated responses to heat stress, RNA sequencing was conducted on nine samples (three biological replicates per group): seedlings subjected to heat stress alone (HS), seedlings pre-treated with melatonin before heat stress (MH), and a control group (CK). The sequencing generated 18.23–23.80 million raw reads per sample, totaling 181.7 million reads, with an average mapping rate of 84.99% to the reference genome [11] (Appendix A). Principal component analysis (PCA) showed clear separation among the three groups, indicating distinct transcriptomic responses, with replicates clustering tightly within each group (Figure 1A).

DEGs were identified using DESeq2 [12] with thresholds of fold change ≥ 2 and false discovery rate (FDR) < 0.01 across three pairwise comparisons: CK vs. HS, CK vs. MH, and HS vs. MH. In total, 3491 (1789 upregulated and 1702 downregulated), 3280 (1688 upregulated and 1592 downregulated), and 1171 (741 upregulated and 430 downregulated) DEGs were identified in these comparisons, respectively (Figure 1B and Appendix A). Venn diagram analysis revealed 19 DEGs common to all comparisons, while 48 genes were consistently induced across CK, HS, and MH treatments. Additionally, 494 DEGs (366 upregulated and 128 downregulated) were unique to MH treatment (Figure 2A,B). Notably, 742, 275, and 356 DEGs were exclusively expressed in CK vs. HS, CK vs. MH, and HS vs. MH comparisons, respectively. Across treatments, upregulated genes generally outnumbered downregulated genes.

### 2.2. Melatonin Exhibits Distinct Heat Resistance Mechanisms

To further elucidate the functional roles of differentially expressed genes (DEGs), Gene Ontology (GO) enrichment analysis was conducted for DEGs identified between treatments, as shown in the dot plot in Figure 2C and detailed in Appendix A. Under heat stress (HS), compared to the control (CK), upregulated DEGs were predominantly enriched in biological processes such as chloroplast RNA processing (GO:0031425), plant-type cell wall biosynthesis (GO:0009595), hydrolase activity (GO:0004553), and cytidine to uridine editing (GO:0016554). Conversely, downregulated DEGs were significantly enriched in pathways related to response to wounding (GO:0009611), jasmonic acid and ethylene-dependent systemic resistance (GO:0009861), and anion transmembrane transporter activity (GO:0008509) (GO:0008509) (Appendix A).

In the CK vs. MH comparison, upregulated DEGs were enriched in processes such as chloroplast RNA processing (GO:0031425) and plant-type cell wall biosynthesis (GO:0009595), as well as responses to chitin (GO:0010200) and water deprivation (GO:0009414). Downregulated DEGs were notably enriched in pathways associated with jasmonic acid and ethylene-dependent systemic resistance (GO:0009861) and response to wounding (GO:0009611).

In contrast, the HS vs. MH comparison revealed distinct GO enrichment patterns. The response to chitin (GO:0010200) emerged as the most significantly enriched GO term among upregulated DEGs. Additionally, unique enrichment was observed in processes such as the regulation of defense responses (GO:0031347), diacylglycerol metabolic process (GO:0046339), and monoacylglycerol metabolic process (GO:0046462). For downregulated DEGs, significant enrichment was identified in the secondary metabolite biosynthetic process (GO:0044550), monooxygenase activity (GO:0004497), and defense response to fungi (GO:0009817).

### 2.3. Melatonin Modulates Multiple Metabolic Pathways Under Heat Stress

To gain deeper insights into the molecular functions of the DEGs, KEGG (Kyoto Encyclopedia of Genes and Genomes) pathway analysis was performed, revealing additional details about the biological roles of these genes (Appendix A). The KEGG enrichment results aligned closely with the GO enrichment findings, highlighting similar pathways enriched in both CK vs. HS and CK vs. MH comparisons. For example, both comparisons showed significant enrichment in protein processing in the endoplasmic reticulum (map04141), glycine, serine, and threonine metabolism (map00260), and zeatin biosynthesis (map00908). Conversely, downregulated genes in these comparisons were enriched in pathways associated with sesquiterpenoid and triterpenoid biosynthesis (map00909), steroid biosynthesis (map00100), and the MAPK signaling pathway—plant (map04016).

Notably, transcripts of nearly all genes involved in the KEGG pathway Zeatin biosynthesis were significantly upregulated in both HS and MH treatments compared to CK (Figure 3C), suggesting that these genes may play a critical role in heat stress responses. This observation underscores the potential importance of the zeatin biosynthesis pathway in enhancing plant resilience to environmental stresses.

In the KEGG enrichment analysis for the HS vs. MH comparison, upregulated DEGs were significantly enriched in pathways such as nicotinate and nicotinamide metabolism (map00760), MAPK signaling (map04016), and purine metabolism (map00760). Meanwhile, downregulated DEGs were associated with pathways including protein processing in the endoplasmic reticulum (map04141), linoleic acid metabolism (map00591), and amino sugar and nucleotide sugar metabolism (map00520).

Notably, genes involved in the nicotinate and nicotinamide metabolism pathway exhibited markedly higher transcript levels in MH-treated samples compared to both CK and HS groups, suggesting a strong association with the application of exogenous melatonin. Furthermore, these genes were identified as forming a distinct gene cluster on chromosome 11, emphasizing their potential regulatory role in melatonin-mediated heat stress tolerance.

### 2.4. K-Means Analysis of DEGs

To further explore the expression patterns of DEGs, we combined the results from the three pairwise comparisons (CK vs. HS, CK vs. MH, HS vs. MH) and retained genes with expression levels greater than 5 in at least one sample. This filtering step identified 3397 DEGs (fold change > 2, adjusted *p*-value ≤ 0.05), which were grouped into eight distinct gene clusters using the k-means clustering algorithm (Appendix A). Among these clusters, clusters 7, 8, and 3 contained genes predominantly expressed in the CK, HS, and MH groups, respectively (Figure 4). Notably, cluster 7, representing CK-specific genes, contained the highest number of genes, highlighting its complexity. Cluster 2 genes showed a gradual increase in expression from CK to HS, whereas cluster 4 genes exhibited an opposing trend, with higher expression in CK and MH. Cluster 1 genes were highly expressed in both HS and MH, while cluster 5 genes displayed a continuous increase in expression from CK to HS to MH.

GO and KEGG enrichment analyses were conducted for each cluster, revealing distinct functional patterns (Figure 4). For instance, cluster 1 was enriched in GO terms related to thylakoid and chloroplast RNA processing, as well as KEGG pathways such as protein processing in the endoplasmic reticulum and glycine, serine, and threonine metabolism. In contrast, cluster 7, which showed an opposite expression pattern to cluster 1, was enriched in GO terms associated with anion transmembrane transporter activity and response to wounding, along with KEGG pathways related to steroid biosynthesis and sesquiterpenoid and triterpenoid biosynthesis.

Clusters 3 and 5 exhibited the highest expression levels in the MH group, suggesting their key roles in melatonin-enhanced heat tolerance in sweet potato. GO analysis revealed that cluster 5 was enriched in terms related to plant-type cell wall biosynthesis and brassinosteroid transferase activity, while cluster 3 was enriched in processes like response to chitin and organonitrogen compounds. KEGG pathway analysis further showed that cluster 5 was associated with defense-related pathways, such as starch and sucrose metabolism and cyanoamino acid metabolism, whereas cluster 3 was enriched in MAPK signaling and plant hormone signal transduction pathways. These findings are consistent with DEG comparisons, underscoring the functional importance of these clusters in the response to heat stress and melatonin treatment.

### 2.5. Quantitative Validation of DEGs Under Heat Stress

To verify the reliability of our transcriptomic data and further investigate the role of exogenous melatonin in enhancing heat tolerance in sweet potato, we conducted qPCR analysis on several genes with higher expression ratios in the HS vs. MH and CK vs. MH groups. The selected genes included WRKY transcription factors, such as *IbWRKY33* (IB06G19780), *IbWRKY7* (IB01G10370), and *IbWRKY40* (IB08G11330); heat shock proteins like *IbHSP21* (IB12G21990); the heat stress-associated gene *IbHSA32* (IB11G34820); and the MAP kinase gene *IbMPK3* (IB03G06030) (Appendix A, Figure 5). The qPCR results revealed expression patterns that were consistent with those observed in the transcriptomic analysis, further corroborating the accuracy and reliability of the RNA-seq data. These findings provide additional validation for the gene expression trends identified in our study and highlight the potential roles of these genes in melatonin-mediated heat stress tolerance.

## 3. Discussion

### 3.1. Exogenous Melatonin and the Antioxidant System in Sweet Potato Leaves

Heat stress induces complex intracellular signaling pathways, primarily involving reactive oxygen species (ROS), which act as mediators in plant stress responses [2]. Exogenous melatonin has been widely recognized for its protective role against various abiotic stresses, including heat stress. Previous studies, including our earlier work, have demonstrated that melatonin pre-treatment alleviated oxidative damage caused by heat stress in sweet potato by reducing oxidative markers such as hydrogen peroxide (H_2_O_2_) and malondialdehyde (MDA) [10]. Heat stress typically increases the levels of oxidative stress markers, such as hydrogen peroxide (H_2_O_2_), superoxide ions, and malondialdehyde (MDA) [10]. However, our previous results demonstrated that melatonin significantly reduced the accumulation of these ROS, suggesting that it enhances the antioxidant capacity of sweet potato leaves [10]. For instance, the receptor-like kinase gene *IbRLK7,* which is known to affect germination speed and oxidant stress in *Arabidopsis* [13], was upregulated in melatonin-treated plants under heat stress. This suggests that melatonin may directly enhance the signaling pathways involved in oxidative stress mitigation. Furthermore, the upregulation of enzymatic antioxidants, such as superoxide dismutase (SOD) and catalase (CAT), such as *IbRGA2*, which may be involved in reducing ROS accumulation in response to stress by up-regulating the transcription of superoxide dismutases [14], aligns with previous reports that melatonin activates the antioxidant defense system to scavenge ROS and stabilize cellular structures [15,16]. These mechanisms likely contribute to the observed improvements in membrane stability and cellular homeostasis under heat stress.

Additionally, melatonin-induced changes in pathways related to chloroplast RNA processing, hydrolase activity, and plant-type cell wall biosynthesis highlight its role in fortifying cellular defense. These processes likely contribute to the mitigating oxidative damage, further underscoring the protective effects of melatonin under heat stress [10,17]. Furthermore, the upregulation of genes involved in responses to chitin and water deprivation suggests that melatonin not only strengthens antioxidant defense but also enhances stress tolerance by modulating metabolic pathways related to cell wall integrity and water homeostasis.

### 3.2. Regulation of Heat Tolerance Through Nicotinate and Nicotinamide Metabolism

The nicotinate and nicotinamide metabolism cycle plays a central role in the biosynthesis of nicotinamide adenine dinucleotide (NAD+), a crucial coenzyme involved in energy metabolism, redox homeostasis, DNA repair, and stress-related protein activation [18,19]. In this study, the nicotinate and nicotinamide metabolism pathways were significantly enriched in melatonin-treated sweet potato plants under heat stress (Figure 3A), with key genes showing high expression levels in MH (Figure 3D). As NAD+ supports antioxidant enzyme activities, its elevated levels have been linked to enhanced stress tolerance, including heat stress [20]. Our findings suggest that melatonin modulates NAD+ biosynthesis, thereby increasing the plant’s resilience to heat-induced oxidative damage. This is in line with studies in other plant species demonstrating that NAD+ biosynthesis contributes tolerance against various abiotic stresses [21,22].

In sweet potato, activation of the nicotinate and nicotinamide metabolism pathway may regulate antioxidant enzyme activities, thereby improving oxidative stress tolerance. Additionally, the upregulation of heat shock proteins (HSPs), such as *IbHSP21*, IbHSP101, and *IbHSP90.1* (Appendix A), which are critical for cellular protection and repair under thermal damage [3,23], suggests that melatonin enhances heat tolerance by promoting cellular repair mechanisms alongside oxidative stress defense.

### 3.3. Chitin as Modulators of Heat Tolerance

Melatonin treatment enhanced the plant’s response to chitin (Figure 2C), a fungal cell wall component that triggers plant immune responses. Chitin recognition activates defense signaling pathways, including MAPK cascades, which are associated with heat stress tolerance [24]. The upregulation of chitin-responsive genes, typically linked to plant defense against pathogens, suggests that melatonin may strengthen shared signaling pathways involved in both biotic and abiotic stress responses. For example, chitin recognition activates MAPK cascades, which are also implicated in the regulation of heat stress tolerance. This activation leads to the production of antioxidant enzymes and osmoprotectants, which mitigate oxidative damage caused by heat stress [25,26]. Thus, the observed upregulation of chitin-responsive genes in melatonin-treated sweet potato highlights a potential cross-talk between immune signaling and heat stress adaptation. This is consistent with studies showing that melatonin-induced defense pathways involving chitin and pathogen-associated molecular patterns (PAMPs) enhance plant resilience under heat stress [27].

### 3.4. Role of Plant Hormones in Melatonin-Mediated Heat Tolerance

Melatonin regulation of cytokinin synthesis, particularly zeatin, plays a critical role in plant stress responses [28]. In our study, KEGG pathway enrichment analysis revealed significant upregulation of genes involved in zeatin biosynthesis in both heat-stressed and melatonin-treated plants (Figure 3A,C). This supports previous findings that melatonin can increase cytokinin levels, thereby enhancing plant resilience to heat stress [29,30]. Cytokinins like zeatin are known to delay senescence, regulate growth, and improve stress tolerance by promoting cell division and activating antioxidant systems [31].

K-means analysis reveals that plant hormone signal transduction pathways were enriched in clusters (clusters 2, 3) containing genes with higher expression levels in HS or MH, indicating the importance of hormonal regulation in heat tolerance. Melatonin treatment modulated the expression of genes involved in the biosynthesis and signaling of key plant hormones, including ethylene, brassinosteroids (BR), and abscisic acid (ABA). These hormones regulate vital processes such as stomatal closure, cell expansion, and the expression of stress-responsive genes [32,33]. Specifically, melatonin upregulated genes associated with brassinosteroid transferase activity, suggesting an enhancement of BR-mediated heat stress tolerance in sweet potato (Figure 4). BRs are known to simultaneously promote growth and stress tolerance [34]. Similarly, ethylene, which regulates cell division and elongation, may synergize with melatonin to improve heat tolerance [32], as indicated by the upregulation of ethylene response factor (ERF) genes such as *IbERF9* (IB08G04660), *IbERF3* (IB13G14360), and *IbERF104* (IB07G01080) in MH (Appendix A). Furthermore, melatonin-induced ABA biosynthesis likely contributes to heat tolerance by reducing water loss through stomatal closure [35,36]. For instance, *IbSCRM,* which encodes an MYC-like bHLH transcriptional activator that binds to the CBF3 promoter in *Arabidopsis*, was induced by melatonin (Appendix A). These findings highlight the central role of plant hormones in mediating heat stress responses and illustrate how melatonin interacts with various hormonal pathways to enhance heat tolerance. This study provides new insights into the regulatory networks orchestrated by melatonin and underscores its potential as a tool for improving plant resilience to heat stress.

## 4. Materials and Methods

### 4.1. Plant Material and Treatments

The sweet potato (*Ipomoea batatas* (L.) Lam.) line used in the experiment was “Haida7791”. The experimental design is described in the published paper by Kumar et al. (2024). In brief, cuttings planted in pots were transferred to a controlled-environment chamber for a 5-day acclimation cultivation (27 °C/24 °C, 4000 Lux, 16/8 h day/night). After that, one group of sweet potato leaves received foliar treatments with 100 mM melatonin, a dose selected based on our previous study, which demonstrated that 100 μM significantly improved heat tolerance in sweet potato compared to other concentrations [10]. The leaves were sprayed every other day for a total of three applications before being subjected to different temperature treatments. The temperature treatments were as follows: control (27 °C/24 °C), HS (42 °C/35 °C), and MH (42 °C/35 °C + 100 μM melatonin). Samples were collected for transcriptome sequencing after 7 days of heat stress treatment. After collection, samples were immediately frozen in liquid nitrogen and stored at −80 °C until RNA extraction.

### 4.2. RNA Extraction and RNA Sequencing

Leaf samples were collected from three treatment groups to investigate the effects of salt stress on gene expression: control (CK), heat stress (HS), and heat stress combined with exogenous melatonin treatment (MH). A total of nine biological replicates (three per group) were obtained. Total RNA was extracted using the RNA Extraction Kit (Vazyme Biotech Co., Ltd., Nanjing, China). RNA concentration, integrity, and yield were assessed using an Agilent 2100 Bioanalyzer (Agilent Technologies, Santa Clara, CA, USA). The extracted RNA was used to synthesize cDNA libraries, where eukaryotic mRNA was enriched using magnetic beads conjugated with Oligo (dT). The cDNA library concentration was also evaluated with the Agilent 2100 Bioanalyzer. RNA sequencing was performed on an Illumina HiSeq X Ten platform (Illumina, San Diego, CA, USA) by Biomarker Technology Company (Beijing, China).

### 4.3. RNA-Seq Data Processing

Paired-end raw reads were subjected to quality control using FastQC [37] (http://www.bioinformatics.babraham.ac.uk/projects/fastqc/, accessed on 10 December 2024). Low-quality bases, including the first 10 nucleotides, were trimmed using the fastx_trim tool from the FASTX-Toolkit [38] (https://github.com/agordon/fastx_toolkit, accessed on 11 December 2024). Clean reads were aligned to the Taizhong 6 genome with the annotation v1.0.a2 [39] using HISAT2 (v2.2.1) [40] with default parameters. Gene-level read counts were calculated using FeatureCount v1.6.4 [41] and normalized to TPM (transcripts per million) for quantifying gene expression. Principal component analysis (PCA) was conducted using log2-transformed TPM values of expressed genes, utilizing the Euclidean distance statistical method. Differential gene expression analysis was performed using the DESeq2 R package (v1.16.1) [12]. For differential expression analysis, the first group in each comparison (e.g., CK in “CK vs. MH”) was used as the baseline to calculate fold changes and identify DEGs.Genes with a false discovery rate (FDR) < 0.01 and |log2-fold change| > 1 were considered differentially expressed genes (DEGs). Enrichment analyses for Gene Ontology (GO) terms and KEGG pathways were performed using the clusterProfiler R package (version “4.4”) [42].

### 4.4. Gene Cluster Analysis

To identify expression patterns, DEGs with TPM values greater than 5 in at least one sample were retained for clustering analysis. Normalized TPM values were used as input for clustering, which was conducted using the visCluster function from the ClusterGVis package (v0.1.0). This approach grouped genes into eight distinct clusters. The clustering and visualization process was completed using ClusterGVis: One-step to Cluster and Visualize Gene Expression Matrix (https://github.com/junjunlab/ClusterGVis, accessed on 23 December 2024).

### 4.5. Expression Analysis by qRT-PCR

Candidate genes identified in the transcriptome analysis were validated using quantitative reverse transcription PCR (qRT-PCR). RNA samples were collected in triplicate to ensure biological reproducibility. Total RNA was extracted with the Plant Tissue RNA Easy Fast RNA Extraction Kit (Tiangen Biotech Co.; Ltd.; Beijing; China). cDNA was synthesized using the TranScript-Uni One-Step gDNA Removal and cDNA Synthesis SuperMix (TransGen Biotech, Beijing, China). qRT-PCR was performed with a SYBR Green RT-PCR Kit (Takara, Dalian, China) using gene-specific primers designed with Primer Premier 5 software (Appendix A). The qRT-PCR program consisted of the following steps: 94 °C for 30 s; 40 cycles of 94 °C for 5 s and 60 °C for 34 s; and a final step at 95 °C for 15 s. The *IbARF* gene served as the internal reference for normalizing expression levels. Each reaction was conducted in triplicate for technical accuracy. Relative expression levels were calculated using the 2^−ΔΔCt^ method.

## 5. Conclusions

This study provides new insights into the molecular and metabolic mechanisms underlying melatonin-induced heat tolerance in sweet potato. Our findings demonstrate that exogenous melatonin effectively mitigates heat stress by enhancing antioxidant defenses, modulating stress-responsive pathways, and regulating key metabolic and hormonal networks. Collectively, our results highlight the multifaceted role of melatonin in mitigating heat stress in sweet potato by orchestrating a complex network of molecular and metabolic processes. This study not only advances our understanding of melatonin’s regulatory functions in plants but also provides a foundation for its application in improving crop resilience to heat stress, which is critical for agriculture in the face of global climate change. Future research could focus on exploring the long-term effects of melatonin treatment under field conditions and elucidating its interactions with other environmental factors. Furthermore, integrating melatonin application with breeding strategies may provide novel approaches for developing heat-tolerant crop varieties.

## Figures and Tables

**Figure 1 plants-14-00430-f001:**
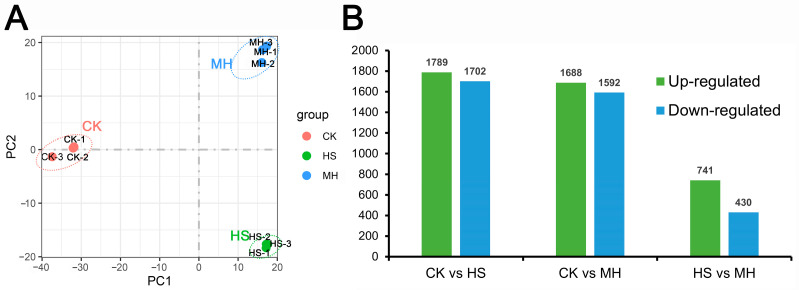
Overview of transcriptome analysis of leaves to CK, HS, and MH treatments. (**A**) Principal component analysis (PCA) plot showing the overall relationships among samples from the three treatments, each with three biological replicates. CK: red; HS: green; MH: blue. Pairwise similarities were measured using the Euclidean distance method on log2-transformed TPM values. (**B**) Number of differentially expressed genes (DEGs) identified in pairwise comparisons between treatments (|fold change| ≥ 1; *p*-value < 0.01).

**Figure 2 plants-14-00430-f002:**
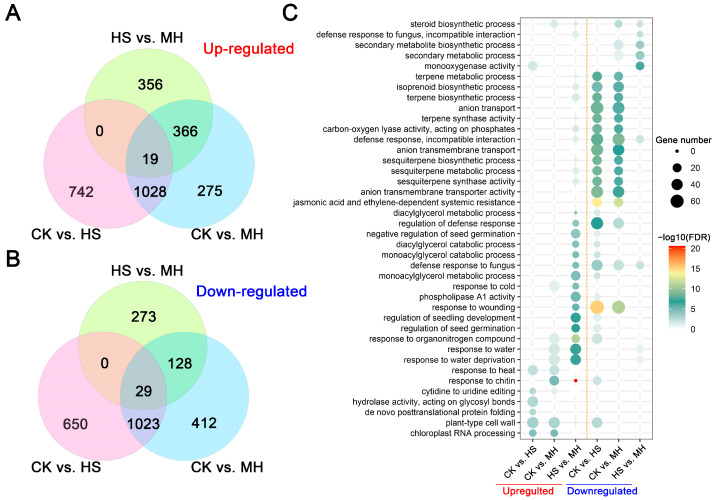
Comparative analysis of the transcriptome for CK, HS, and MH treatment. (**A**,**B**) Venn diagrams showing the overlaps of upregulated (**A**) and downregulated (**B**) DEGs in pairwise comparisons between treatments. (**C**) Enriched Gene Ontology (GO) terms for DEGs in pairwise comparisons. The color and size of the dots represent enrichment significance and the number of associated genes, respectively.

**Figure 3 plants-14-00430-f003:**
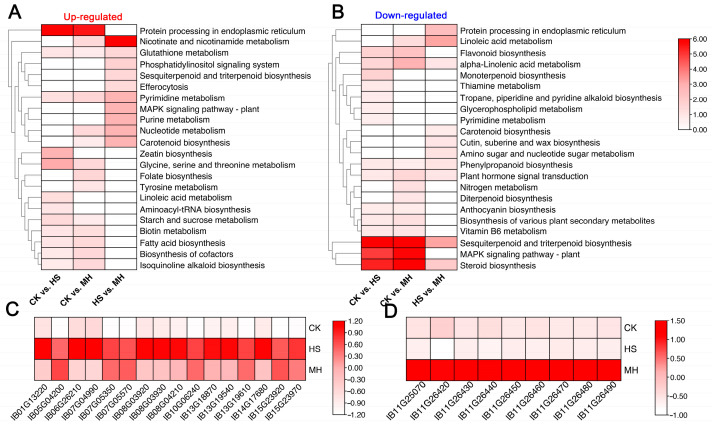
KEGG enrichment analysis of DEGs in pairwise comparison. (**A**) KEGG enrichment analysis of up-regulated DEGs across pairwise comparisons. (**B**) KEGG enrichment analysis of down-regulated DEGs across pairwise comparisons. (**C**) Heatmap showing the RNA profiles of genes involved in the enriched KEGG pathway “Zeatin biosynthesis” across the CK, HS, and MH treatment. (**D**) Heatmap showing the RNA expression profiles of genes involved in the enriched KEGG pathway “Nicotinate and nicotinamide metabolism” across CK, HS, and MH treatments. The color bar represents the Z-scores after normalization.

**Figure 4 plants-14-00430-f004:**
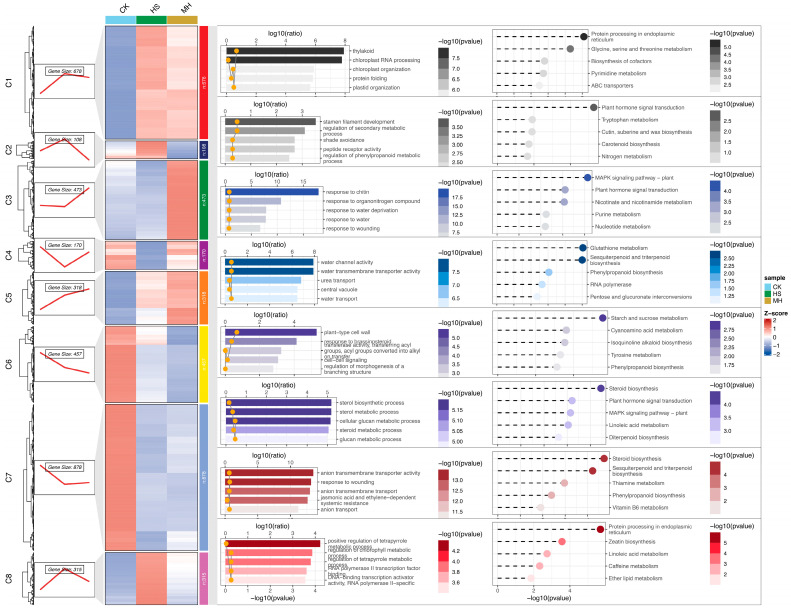
Clustering and functional enrichment analysis of DEGs under different treatments. The heatmap shows the Z-scores of DEGs across CK (control), HS (heat stress), and MH (melatonin under heat stress) treatments. DEGs are grouped into eight clusters (C1–C8) based on expression patterns, with the number of genes in each cluster indicated. Line plots on the left depict the expression trends for each cluster. Functional enrichment analysis for each cluster is presented on the right, with bar charts showing the log10(ratio) of enriched GO terms and KEGG pathways, while bubble plots represent the corresponding −log10(*p*-value) for significant terms. Key biological processes and pathways relevant to each cluster are highlighted, reflecting their functional significance under the treatments.

**Figure 5 plants-14-00430-f005:**
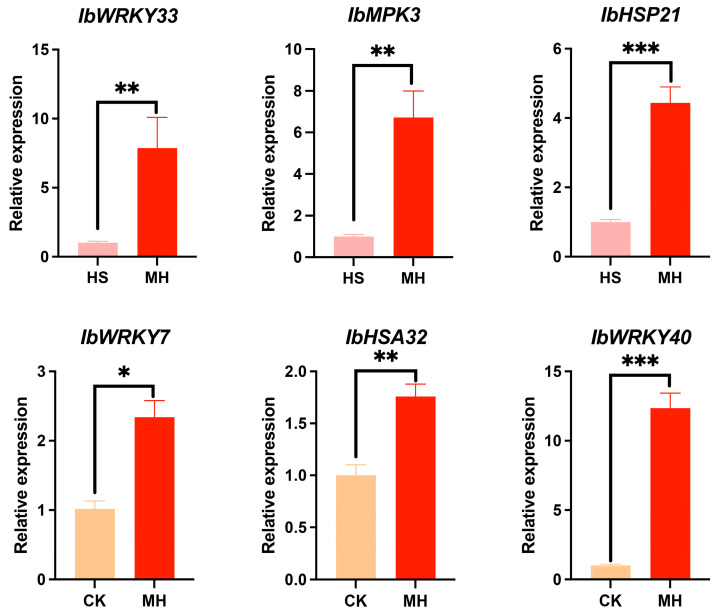
qRT-PCR validation of genes associated with melatonin treatment under heat stress. Error bars represent the means ± SE of three biological replicates. Gene expression levels are presented as the mean of three replicates. Statistical significance was analyzed, with “*” indicating *p* ≤ 0.05, “**” denoting *p* ≤ 0.01, and “***” indicating *p* ≤ 0.001. CK represents the control group, HS refers to heat stress treatment, and MH corresponds to heat stress combined with exogenous melatonin treatment.

## Data Availability

Raw reads of all Illumina RNA-seq library generated in this study are available from the SRA at NCBI (https://www.ncbi.nlm.nih.gov/sra) under the accession number PRJNA1209623.

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
