# Peer review of "Melatonin-Induced Transcriptome Variation of Sweet Potato Under Heat Stress"

_plants, 2025, doi:10.3390/plants14030430_

Round 1
Reviewer 1 Report
Comments and Suggestions for Authors
Manuscript ID: plants-3452517
Title of the manuscript: Melatonin-induced Transcriptome Variation of Sweet Potato Under Heat Stress
In the study, information about “Melatonin-induced Transcriptome Variation of Sweet Potato Under Heat Stress” is provide. I have completed the evaluation of the research. The manuscript is generally well written, detailed and contains original findings. There are still some points in the manuscript that need to be improved. In particular, the research question and hypothesis of the study must be included. I think the research will interest the reader. But after MINOR corrections, the manuscript can be accepted for publication in PLANTS.
Note: My suggestions were shown on PDF file.
With my best regards

Author Response
Line 76: missing research question(s) and hypothesis of the study Pls, add
Response: Thank you for pointing out the absence of explicitly stated research questions and hypotheses in the manuscript. We agree that including these elements will strengthen the clarity and focus of the study. We have revised this section to include the specific research questions and hypotheses guiding this work (Line 81-88).
Line 237: make the discussion more detailed for each parameter. Avoid descriptive statements.
Response: Thank you for your valuable feedback regarding the discussion section. We acknowledge that some parts of the discussion rely heavily on descriptive statements. To address this, we have revised the discussion section to provide a more detailed analysis for each parameter, focusing on the underlying mechanisms and their implications rather than simply describing the results. Hope this will help to deepen the interpretation and make the discussion more insightful.
Line 317: Pls, provide latin name as patato (latin name).
Response: Thank you for your suggestion to include the Latin name for sweet potato. We agree that providing the Latin name would enhance clarity and scientific precision. We have revised the text to include the Latin name.
Line 321: Why did you use this dose? please add reference
Response: Thank you for your question regarding the selection of the melatonin dose. The dose of 100 μM was chosen based on the results of our prior physiological experiments, published in Horticultural Plant Journal (Kumar et al., 2024). In our previous study, we tested various concentrations of melatonin (0, 10, 25, 50, and 100 μM) as pre-treatments. The results showed that a concentration of 100 μM significantly enhanced heat tolerance in sweet potato compared to other doses. Therefore, we selected 100 μM as the optimal concentration for further transcriptome analysis in this study. We have added this reference and explanation to the manuscript in section 4.1.
Reviewer 2 Report
Comments and Suggestions for Authors
Title: OK
Abstract: OK
Introduction
Lines 38-42: include citation for this affirmation
Results
Line 86: 90.99% mapping rate. I checked Table S1 and the average of uniquely map reads gives me 84.99%.
Figure 1.(A): This looks to be a PCA plot, not an MDS plot. If you used PCA, please also change in MyM
Lines 90-91:“DEGs were identified using DESeq2 [11] with thresholds of fold change ≥2 and false 90 discovery rate (FDR) < 0.01” is already mentioned in MyM, but please check if you are using fold change "2" or "1" as there is a discrepancy in the text, I think it is "1".
Line 94: respectively (Fig. 2B 1B and Fig. S2)
Line 120: Could you clarify whether the differential expression analysis was performed as 'CK vs MH' or 'MH vs CK'? The interpretation depends critically on the comparison order. The same question is for others comparisons.
Lines 140-179: you forgot to mentioned Table S4
Line 177: K-menas means analysis of DEGs
Line 285: delay delay, senescence, regulate growth, and improve
Discussion
Line 246: denonstrated demonstrated
Materials and methods
Line 325: Samples were stored in liquid nitrogen? Please indicate that
Lines 339: correct citation could be
Andrews, S. (2017). FastQC: a quality control tool for high throughput sequence data Available online: http://www.bioinformatics.babraham.ac.uk/projects/fastqc/ (accessed on 9 Septembre 2024)
Lines 341: correct citation could be
Hannon, G. J. (2010). FASTX-Toolkit: FASTQ/A short-reads pre-processing tools. Available online: http://hannonlab.cshl.edu/fastx_toolkit (accessed on 9 Septembre 2024)
Author Response
Lines 38-42: include citation for this affirmation
Response: Thank you for pointing out the need for proper citations in Lines 38–42 to support the affirmation. We have included appropriate references to strengthen the statement regarding the impact of heat stress on plant growth and development.
Results
Line 86: 90.99% mapping rate. I checked Table S1 and the average of uniquely map reads gives me 84.99%.
Response: Thank you for pointing out the discrepancy regarding the reported mapping rate. Upon re-examining the data in Table S1, you are correct that the average unique mapping rate is 84.99%, not 90.99% as stated in the main text. We appreciate your attention to this detail and we have revised the manuscript to reflect the correct value (Line 104).
Figure 1. (A): This looks to be a PCA plot, not an MDS plot. If you used PCA, please also change in MyM
Response: Thank you for catching this error in the description of Figure 1(A). You are correct that the analysis used Principal Component Analysis (PCA) rather than Multi-Dimensional Scaling (MDS). We apologize for the mislabeling and we have revised the manuscript to reflect the correct method in both the figure legend and the main text.
Lines 90-91:“DEGs were identified using DESeq2 [11] with thresholds of fold change ≥2 and false 90 discovery rate (FDR) < 0.01” is already mentioned in MyM, but please check if you are using fold change "2" or "1" as there is a discrepancy in the text, I think it is "1".
Response: Thank you for your careful review and for pointing out this potential discrepancy. Upon verification, the use of "fold change ≥ 2" in the main text and "log2 fold change ≥ 1" in the Methods section is correct and consistent. The threshold of "fold change ≥ 2" in the main text is equivalent to "log2 fold change ≥ 1," as the latter refers to the logarithmic transformation of the fold change values.
Line 94: respectively (Fig. 2B 1B and Fig. S2)
Response: Corrected.
Line 120: Could you clarify whether the differential expression analysis was performed as 'CK vs MH' or 'MH vs CK'? The interpretation depends critically on the comparison order. The same question is for others comparisons.
Response: Thank you for pointing this out. We apologize for any confusion caused by the inconsistent labeling of differential expression comparisons. The comparison order was indeed ‘CK vs MH’, where CK (control) is used as the baseline to compare the expression changes in the MH (melatonin treatment under heat stress) group. This comparison order is consistent throughout the study. Similarly, the other comparisons, ‘CK vs HS’ and ‘HS vs MH’, also follow the same logic, where the first group serves as the baseline for comparison. We have revised the section 4.3 to make this clear.
Lines 140-179: you forgot to mentioned Table S4
Response: Thank you for pointing this out. We have revised the manuscript and added a reference to Table S4 in section 2.3.
Line 177: K-menas means analysis of DEGs
Response: Thank you for identifying this typographical error. We have corrected "K-menas" to "k-means" to ensure accuracy and clarity in describing the clustering analysis.
Line 285: delay delay, senescence, regulate growth, and improve
Response: Thank you for catching this typographical error. We have corrected the redundancy in the phrase to ensure clarity and readability.
Discussion
Line 246: denonstrated demonstrated
Response: Corrected.
Materials and methods
Line 325: Samples were stored in liquid nitrogen? Please indicate that
Response: Thank you for pointing this out. We agree that providing this information is important for ensuring clarity and reproducibility. We have revised the text to explicitly state that samples were immediately frozen in liquid nitrogen and stored at -80°C until further processing.
Lines 339: correct citation could be
Andrews, S. (2017). FastQC: a quality control tool for high throughput sequence data Available online: http://www.bioinformatics.babraham.ac.uk/projects/fastqc/ (accessed on 9 Septembre 2024)
Response: Thank you for your suggestion. We have updated the citation for FastQC to reflect the correct format as provided. This ensures accuracy and proper attribution for the tool used in our study.
Lines 341: correct citation could be
Hannon, G. J. (2010). FASTX-Toolkit: FASTQ/A short-reads pre-processing tools. Available online: http://hannonlab.cshl.edu/fastx_toolkit(accessed on 9 Septembre 2024)
Response: Thank you for your helpful suggestion. We have updated the citation for the FASTX-Toolkit to the correct format as recommended.
Reviewer 3 Report
Comments and Suggestions for Authors
The reviewed article presents results of transcriptomic analysis of melatonin-treated sweet potato plants under heat stress. This approach was used to identify the mechanisms of melatonin-induced plant defense against heat stress. The results are interesting and mostly well described and discussed, but I have some recommendations for improving the text.
1. Abstract. The abbreviations ERF and BEE2 should be deciphered. I found the definition of ERF below in the text, but BEE2 is not mentioned anywhere else in the text. BEE2 is a rarely used term. Does it mean brassinolide enhanced expression (component of brassinolide signaling)? Why is not this gene mentioned anywhere else in the text?
2. Lines 43-44. “These stress-induced changes trigger complex protective responses in plants, including the upregulation of heat shock proteins, synthesis of other stress-related proteins, and production of reactive oxygen species (ROS)” - production of reactive oxygen species plays a dual role in stressed plants. Although ROS can trigger protective responses, high concentrations are detrimental to stressed plants. This must be specified. Otherwise, the phrase about detoxification of ROS by melatonin looks counterintuitive.
3. Lines 56-59. “Melatonin’s involvement in plant stress tolerance extends beyond ROS scavenging. It has been shown to regulate the expression of heat shock proteins, activate antioxidant enzymes, and stabilize cell membranes under heat stress conditions [5].” – but stabilization of cell membranes is indicator ROS scavenging and so it does not extend beyond ROS scavenging.
4. The title “2.3. Nicotinate and nicotinamide metabolism pathway gene are upregulated by melatonin” should be modified. This section provides information on many effects beyond the metabolism of nicotinate and nicotinamide.
5. Abbreviation KEGG should be deciphered and its principle shortly described. Does it mean Kyoto Encyclopedia of Genes and Genomes?
6. “K-menas analysis” – is this correct? Below the k-means clustering algorithm is mentioned.
7. Lines 241-242. “In this study, melatonin pre-treatment alleviated oxidative damage caused by heat stress in sweet potato, consistent with previous findings…”. Since there is no literature reference in this sentence, it can be assumed that the results of the present study are being discussed. But no results of measuring indicators of oxidative damage can be found in this article. So it should be clarified that literature data are discussed and reference should be provided.
8. Lines 245-247. “However, our results demonstrated that melatonin significantly reduced the accumulation of these ROS, suggesting that it enhances the antioxidant capacity of sweet potato leaves.” – similar comment. No data on ROS accumulation were provided in the result section of the present article.
9. Lines 289-291. “The upregulation of chitin-responsive genes in melatonin-treated sweet potato suggests that melatonin strengthens innate immune responses, contributing to improved heat stress tolerance.” – It not clear for me how innate immune responses is associated with improved heat stress tolerance.
10. Information about genes involved in zeatin synthesis should be moved to section 3.4, since zeatin is hormone.
11. Line 307-309. “Furthermore, melatonin-induced ABA biosynthesis likely contributes to heat tolerance by reducing water loss through stomatal closure” – There is no information about ABA in Results section. I found information about abscisic acid related genes is Suppl. 5 table. I think authors should at least make a reference to it in regard to ABA.
Author Response
- Abstract. The abbreviations ERF and BEE2 should be deciphered. I found the definition of ERF below in the text, but BEE2 is not mentioned anywhere else in the text. BEE2 is a rarely used term. Does it mean brassinolide enhanced expression (component of brassinolide signaling)? Why is not this gene mentioned anywhere else in the text?
Response: Thank you for your valuable feedback. We appreciate your careful review and attention to detail. To address your concerns: The abbreviation ERF (ethylene response factor) has now been defined in the abstract for clarity. Regarding BEE2, this gene is indeed part of the brassinosteroid signaling pathway, specifically referring to brassinosteroid enhanced expression 2. However, as BEE2 is not discussed further in the main text or results, we acknowledge that its inclusion in the abstract may cause confusion. We have removed BEE2 from the abstract to maintain focus on genes that are thoroughly discussed in the manuscript.
- Lines 43-44. “These stress-induced changes trigger complex protective responses in plants, including the upregulation of heat shock proteins, synthesis of other stress-related proteins, and production of reactive oxygen species (ROS)” - production of reactive oxygen species plays a dual role in stressed plants. Although ROS can trigger protective responses, high concentrations are detrimental to stressed plants. This must be specified. Otherwise, the phrase about detoxification of ROS by melatonin looks counterintuitive.
Response: Thank you for your insightful comment. We appreciate your observation regarding the dual role of ROS in plants under stress. To address this, we have revised the text to specify the dual role of ROS, highlighting their signaling function at low concentrations and their detrimental effects at high concentrations. This clarification ensures consistency with subsequent discussions about the detoxification of ROS by melatonin.
- Lines 56-59. “Melatonin’s involvement in plant stress tolerance extends beyond ROS scavenging. It has been shown to regulate the expression of heat shock proteins, activate antioxidant enzymes, and stabilize cell membranes under heat stress conditions [5].” – but stabilization of cell membranes is indicator ROS scavenging and so it does not extend beyond ROS scavenging.
Response: Thank you for your comment. We agree that the stabilization of cell membranes is closely linked to the effects of ROS scavenging and should not be presented as an independent function beyond ROS scavenging. To address this, we have revised the statement to clarify the relationship between these processes and better reflect the role of melatonin (Line 62-66).
- The title “2.3. Nicotinate and nicotinamide metabolism pathway gene are upregulated by melatonin” should be modified. This section provides information on many effects beyond the metabolism of nicotinate and nicotinamide.
Response: Thank you for this valuable suggestion. We agree that the current title does not fully capture the broader range of effects discussed in this section. To better reflect the content, we have revised the title to encompass the multiple metabolic pathways influenced by melatonin.
- Abbreviation KEGG should be deciphered and its principle shortly described. Does it mean Kyoto Encyclopedia of Genes and Genomes?
Response: Thank you for pointing this out. We agree that deciphering KEGG and briefly describing its principle would enhance the clarity of the manuscript. We have revised the text to include the full form of KEGG (Kyoto Encyclopedia of Genes and Genomes) and added a concise explanation of its role in functional analysis in section 2.3.
- “K-menas analysis” – is this correct? Below the k-means clustering algorithm is mentioned.
Response: Thank you for pointing out this typographical error. You are correct that “K-menas analysis” was a mistake, and it should be “k-means clustering algorithm.” We have corrected it in line 204.
- Lines 241-242. “In this study, melatonin pre-treatment alleviated oxidative damage caused by heat stress in sweet potato, consistent with previous findings…”. Since there is no literature reference in this sentence, it can be assumed that the results of the present study are being discussed. But no results of measuring indicators of oxidative damage can be found in this article. So it should be clarified that literature data are discussed and reference should be provided.
Response: Thank you for highlighting this point. We acknowledge that this statement could be misinterpreted as being based on the present study's results, even though it was intended to reference previously published findings. To clarify, we have revised the text to explicitly state that this conclusion is supported by prior study, and we have added the appropriate reference to our earlier work (Kumar et al., 2024) (Line 270-273).
- Lines 245-247. “However, our results demonstrated that melatonin significantly reduced the accumulation of these ROS, suggesting that it enhances the antioxidant capacity of sweet potato leaves.” – similar comment. No data on ROS accumulation were provided in the result section of the present article.
Response: Thank you for your insightful comment. We appreciate your observation regarding the lack of direct ROS accumulation data in the current manuscript. To address this, we have revised the text to clarify that the conclusion regarding reduced ROS levels is based on previously published results (Kumar et al., 2024). In this earlier study, we reported that melatonin significantly reduced ROS markers such as hydrogen peroxide (H₂O₂) and malondialdehyde (MDA) in heat-stressed sweet potato plants. We have now explicitly referenced this prior publication to support the statement.
- Lines 289-291. “The upregulation of chitin-responsive genes in melatonin-treated sweet potato suggests that melatonin strengthens innate immune responses, contributing to improved heat stress tolerance.” – It not clear for me how innate immune responses is associated with improved heat stress tolerance.
Response: Thank you for pointing out the need to clarify the connection between innate immune responses and heat stress tolerance. We have revised this section to provide a more detailed explanation. Specifically, we have elaborated on how the upregulation of chitin-responsive genes, which are commonly associated with plant defense against biotic stress, can also enhance abiotic stress tolerance by activating shared downstream signaling pathways such as MAPK cascades and the production of reactive oxygen species (ROS) scavengers (Line 329-334).
- Information about genes involved in zeatin synthesis should be moved to section 3.4, since zeatin is hormone.
Response: Thank you for your valuable suggestion. We agree that the discussion on genes involved in zeatin synthesis aligns more appropriately with section 3.4, which focuses on the role of plant hormones in heat tolerance. We have moved the zeatin synthesis section to section 3.4.
- Line 307-309. “Furthermore, melatonin-induced ABA biosynthesis likely contributes to heat tolerance by reducing water loss through stomatal closure” – There is no information about ABA in Results section. I found information about abscisic acid related genes is Suppl. 5 table. I think authors should at least make a reference to it in regard to ABA.
Response: Thank you for pointing this out. We appreciate your observation regarding the absence of explicit results about ABA in the main text. To address this, we have added a gene (IbSCRM) in this part to highlight the presence of ABA-related genes and clarified the connection between ABA biosynthesis and heat tolerance.
Round 2
Reviewer 1 Report
Comments and Suggestions for Authors
Dear Editor Ms. Lea Tao
Manuscript ID: plants-3452517
Title of the manuscript: Melatonin-induced Transcriptome Variation of Sweet Potato Under Heat Stress
I suggested some corrections to improve the manuscript. I saw that these suggestions were implemented point by point. The manuscript was improved. Now, the manuscript can be accepted for publication in PLANTS.
With my best regards
Reviewer 2 Report
Comments and Suggestions for Authors
You're welcome!
Congratulations on your publication, very interesting for the scientific community.
Reviewer 3 Report
Comments and Suggestions for Authors
Authors carefully addressed my recommendations and revised the text in accordance with all my comments. I am satisfied with the modifications and think that this article may be published in its present form